# Heme Oxygenase-1 Exerts Antiviral Activity against Hepatitis A Virus In Vitro

**DOI:** 10.3390/pharmaceutics13081229

**Published:** 2021-08-09

**Authors:** Dong-Hwi Kim, Hee-Seop Ahn, Hyeon-Jeong Go, Da-Yoon Kim, Jae-Hyeong Kim, Joong-Bok Lee, Seung-Yong Park, Chang-Seon Song, Sang-Won Lee, In-Soo Choi

**Affiliations:** Department of Infectious Diseases, College of Veterinary Medicine, Konkuk University, 120 Neungdong-ro, Gwangjin-gu, Seoul 05029, Korea; opeean0@naver.com (D.-H.K.); heesuob2@naver.com (H.-S.A.); misilseju@naver.com (H.-J.G.); kimdayoon6@naver.com (D.-Y.K.); bluetown30@naver.com (J.-H.K.); virus@konkuk.ac.kr (J.-B.L.); paseyo@konkuk.ac.kr (S.-Y.P.); songcs@konkuk.ac.kr (C.-S.S.); odssey@konkuk.ac.kr (S.-W.L.)

**Keywords:** hepatitis A virus, heme oxygenase-1, hemin, antiviral therapeutics

## Abstract

Hepatitis A virus (HAV), the causative pathogen of hepatitis A, induces severe acute liver injuries in humans and is a serious public health concern worldwide. However, appropriate therapeutics have not yet been developed. The enzyme heme oxygenase-1 (HO-1) exerts antiviral activities in cells infected with several viruses including hepatitis B and C viruses. In this study, we demonstrated for the first time the suppression of virus replication by HO-1 in cells infected with HAV. Hemin (HO-1 inducer) induced HO-1 mRNA and protein expression, as expected, and below 50 mM, dose-dependently reduced the viral RNA and proteins in the HAV-infected cells without cytotoxicity. Additionally, HO-1 protein overexpression using a protein expression vector suppressed HAV replication. Although ZnPP-9, an HO-1 inhibitor, did not affect HAV replication, it significantly inhibited hemin-induced antiviral activity in HAV-infected cells. Additionally, FeCl_3_, CORM-3, biliverdin, and the HO-1 inducers andrographolide and CoPP inhibited HAV replication in the HAV-infected cells; andrographolide and CoPP exhibited a dose-dependent effect. In conclusion, these results suggest that HO-1 effectively suppresses HAV infection in vitro, and its enzymatic products appear to exert antiviral activity. We expect that these results could contribute to the development of a new antiviral drug for HAV.

## 1. Introduction

Hepatitis A, caused by infection with hepatitis A virus (HAV), presents clinical signs such as jaundice, fatigue, and abdominal pains. HAV, a positive sense single-stranded RNA virus with a 7.5 kb genome, is classified into the genus *Hepatovirus* of the family *Picornaviridae* [1]. HAV was originally classified as a non-enveloped virus; however, some studies revealed that it can exist in an enveloped form by hijacking cellular membranes [2,3]. Typical transmission of HAV occurs by direct contact with patients or ingestion of contaminated foods or water through the fecal–oral route [4,5,6]. HAV causes only acute hepatitis, not chronic hepatitis [7]. The degree of symptoms is more likely to be severe in older patients or patients with chronic liver diseases [8,9].

Several vaccines have been developed since HAV was solely isolated in 1979. An inactivated vaccine and a live attenuated HAV vaccine are currently available [10]. Despite the development of effective vaccines, hepatitis A remains the leading disease of food-mediated viral infection, resulting in significant economic losses and health problems worldwide [11,12,13]. Recently, epidemiological changes in HAV infection had been observed in the Unites States [14]. In 2015 alone, approximately 11,000 patients died worldwide from hepatitis A [15]. Once infected with HAV, recovery from the disease takes weeks or months. Although anti-viral therapeutics could attenuate the pathogenicity of the virus and limit epidemiological outbreaks, therapeutic drugs specific for HAV have not been developed to date because of the reliance on vaccines developed in the early 1990s.

Heme oxygenase-1 (HO-1), a stress-inducible heat shock protein (HSP), is an anti-inflammatory and anti-oxidative protein identified as HSP32. It catalyzes the breakdown of heme to carbon monoxide (CO), ferrous iron (Fe^2+^), and biliverdin [16]. Originally, HO-1 was known to be induced in stressful conditions such as oxidative stress, hypoxia, heat shock, and exposure to heavy metals and cytokines in the biological system [17,18,19]. It has been attracting attention as an enzyme that acts as a therapeutic by protecting cells from these stressors [20]. Recently, several studies revealed that HO-1 exerts direct antiviral activities against hepatitis B virus (HBV), hepatitis C virus (HCV), and dengue virus (DENV) infection and an indirect effect against influenza [21]. Additionally, it suppresses the transcription and replication of Ebola virus; however, the exact mechanism remains unknown [22].

Hemin is a degradation product molecule of hemoglobin, which plays an integral role in oxygen transport, mitochondrial function, and various signal transduction routes. It induces HO-1 by separating the nuclear factor E2-related factor 2 (Nrf2) from Kelch-like ECH-associated protein 1 (Keap1). The separated Nrf2 is phosphorylated and translocated into the nucleus. Subsequently, the transcription regulator protein BACH1-transcription factor musculoaponeurotic fibrosarcoma (MAF) complex, which suppresses the expression of HO-1, is dissociated, releasing MAF, which binds with Nrf2 and induces HMOX1 transcription [21,23].

In this study, hemin was used as an HO-1 inducer to examine the antiviral effect of HO-1 on HAV infection in vitro. Induction and expression of HO-1 effectively inhibited HAV, which implies that a novel and effective strategy could be developed against HAV infection. This is the first study to reveal that HO-1 efficiently inhibits HAV replication.

## 2. Materials and Methods

### 2.1. Cell Lines, Viruses, and Chemicals

The fetal rhesus kidney (FRhK-4) cell is a HAV-permissive cell line derived from the rhesus monkey. Dulbecco’s modified Eagle’s medium (DMEM) containing 4% heat-inactivated fetal bovine serum (FBS, Gibco, Campinas, Brazil) and antibiotic-antimycotic reagent (Gibco, Campinas, Brazil) was used to maintain this cell line under a 5% CO_2_ atmosphere at 37 °C.

The HAV strain HM-175/18f, clone B (VR-1402) was obtained from the American Type Culture Collection. Ten genomic equivalents (GE) copies of the HAV per cell were used to inoculate FRhK-4. Cells inoculated with the virus were maintained in DMEM supplemented with 4% heat-inactivated FBS and antibiotic–antimycotic reagent.

We purchased hemin, cobalt protoporphyrin IX (CoPP-9), andrographolide, zinc protoporphyrin IX (ZnPP-9), iron (III) chloride (FeCl_3_), and carbon monoxide-releasing molecule (CORM-3) from Sigma-Aldrich (St. Louis, MO, USA). Biliverdin was purchased from Cayman Chemical (Ann Arbor, MI, USA).

### 2.2. Cell Cytotoxicity Assay

A 3-(4, 5-Dimethylthiazolyl-2)-2, 5-diphenyltetrazolium bromide (MTT) assay was conducted to evaluate the cytotoxicity of hemin, FeCl_3_, CORM-3, and biliverdin against FRhK-4 cells (1 × 10^4^ cells/well) in a 96-well plate. Each of the 4 chemicals was added at specific concentrations 24 h after seeding of the cells and maintained for 72 h. After removing the supernatants, MTT (5 mg/mL), and fresh serum-free media were added and incubated in a 5% CO_2_ atmosphere at 37 °C. After 3 h, 150 μL of dimethyl sulfoxide was added as a solubilization solution. We determined the viable cells by analyzing the absorbance at a wavelength of 540 nm wavelengths using a plate-reading spectrophotometer. Cytotoxicity was calculated as the 50% cytotoxicity concentration (CC_50_) value by using untreated cells as the 100% viability control.

### 2.3. Reverse Transcription-Quantitative Polymerase Chain Reaction

Viral RNA and intracellular mRNA were extracted from the cell lysates using the RNeasy Mini, RNA isolation kit (Qiagen, Hilden, Germany) according to the manufacturer’s instructions. PCR primers and probe sequences for detection of HAV non-coding region were adapted from a study as the followings: forward primer 5′-TTT CCG GAG TCC CTC TTG-3′, reverse primer 5′-AAA GGG AAA ATT TAG CCT ATA GCC-3, and probe primer 5′ FAM-ACT TGA TAC CTC ACC GCC GTT TGC CT-TAMRA 3′ [24]. PCR primers for detection of HAV RNA-dependent RNA polymerase (RdRp) gene were designed by our laboratory: forward primer 5′-TCC TGC AGC TAT GCC CTT TT-3′ and reverse primer 5′- GCC TCT TTA TAA CCC TCT GG -3′. PCR primers for detection of monkey HO-1 and GAPDH were adapted from other studies: HO-1 forward primer 5′-CTT CAA GCT GGT GAT GGC-3′ and reverse primer 5′-TGG AGC CGC TTC ACA TAG-3′ [25]; GAPDH forward primer 5′-GAA ATC CCA TCA CCA TCT TCC AGG-3′ and reverse primer 5′-GAG CCC CAG CCT TCT CCA TG-3′ [26]. We performed the reverse transcription-quantitative polymerase chain reaction (RT-qPCR) for detection of the non-coding region of HAV genome with an AgPath-ID™ One-step RT-PCR kit (Applied Biosystems, Courtaboeuf, France) using Smart Cycler (Cepheid, Sunnyvale, CA, USA) according to the manufacturer’s instructions; 45 °C for 10 min, 95 °C for 15 min, and 40 cycles of 95 °C for 10 s, and 60 °C for 1 min. The viral GE copy numbers were calculated from a standard curve generated using plasmid DNA containing the target sequence of HAV (22–108 bp of HM-175 strain, GenBank No. M14707). We divided the viral genomic copy numbers by 10^8^ GAPDH to calculate the number of viruses contained per cell. The qPCR reaction for RdRp region of HAV and intracellular mRNA were performed with a One Step TB Green^®^ PrimeScript™ RT-PCR Kit (Takara, Tokyo, Japan) using Light Cycler (Roche, Mannheim, Germany) according to the manufacturer’s instructions; 42 °C for 5 min, 95 °C for 10 s, 40 cycles of 95 °C for 5 s and 60 °C for 34 s, 95 °C for 15 s, 60 °C for 1 min, and 95 °C for 15 s. All experiments were repeated three times to get reliable results.

### 2.4. Plasmid Construction and Transfection

The coding sequence of the monkey HO-1 gene (XM028827760) was synthesized and ligated into a 3.1 (+) vector (Invitrogen, Carlsbad, CA, USA). pcDNA3.1 (+) (pcDNA3.1 (+)/mkHO-1) inserted with the monkey HO-1 gene or the empty vector control pcDNA3.1 (+) was subsequently transfected into FRhK-4 cells using GenePORTER 3000 (Genlantis, San Diego, CA, USA). We incubated 70% confluent FRhK-4 cells in a 6-well plate with a mixture of 4 μg DNA vector and GenePORTER 3000 reagent in serum-free medium for 4 h. Following incubation, FBS was added to the medium at 10%.

### 2.5. Western Blot Analysis

The cells were washed 3 times with PBS and collected. Cell lysis was performed in 2× Laemmli sample buffer (S3401, Sigma, St. Louis, MO, USA). The lysates were boiled at 95 °C and centrifuged (9000× *g* for 1 min), and the supernatants were loaded on to SDS-polyacrylamide gel. After electrophoresis, the proteins were transferred to a nitrocellulose membrane and blocked at 4 °C overnight with 5% skim milk in PBS solution containing 0.05% Tween 20 (PBS-T). The membranes were incubated for 1 h at 25 °C with an anti-HO-1 antibody, or an anti-GAPDH antibody (orb397248, Byorbyt, Cambridge, UK), or an anti-HAV VP1 antibody (ABIN2143092, Antibodies-online GmbH, Aachen, Germany) in 2.5% skim milk/PBS-T. Next, the membranes were washed 3 times with PBS-T for 10 min, incubated with a secondary antibody tagged with horseradish peroxidase in 2.5% skim milk/PBS-T solution for 1 h, and washed 3 times with PBS-T for 10 min.

### 2.6. Statistical Analysis

All experiments were conducted at least three times and expressed as the mean ± SD. Student’s *t*-test was performed using PRISM 8.0.1 (Graphpad Software, San Diego, CA, USA); *p* < 0.05 were considered as significant results.

## 3. Results

### 3.1. Inhibition of HAV Replication by Hemin-Mediated Upregulation of HO-1

FRhK-4 cells infected with HAV were treated with different doses of hemin. The mRNA expression levels of HO-1 were determined relative to GAPDH expression. Hemin induced mRNA expression of HO-1 in a dose-dependent manner as expected. RT-qPCR for detection of the non-coding region of HAV RNA was conducted with the same samples to determine the correlation between the quantity of HO-1 mRNA and HAV RNA. As the concentrations of hemin increased, the amount of HAV RNA decreased in the HAV-infected cells (Figure 1A). We additionally verified the reduced expression of the RdRp gene, which plays a key role in the replication of virus (Appendix A). A cell cytotoxicity assay using MTT was conducted because the decrease in the HAV RNA quantity could be reduced by cytotoxicity, which may be induced by high concentrations of hemin. Cell viability was calculated 3 days later in FRhK-4 cells, which were treated with various concentrations of hemin (Figure 1B). The CC_50_ and 50% effective concentration (EC_50_) of hemin were calculated (Table 1). The absence of cell cytotoxicity and significant suppression of the viral RNA replication were confirmed with 50 μM hemin (Figure 1A,B). Therefore, we conducted other experiments with less than 50 μM hemin in this study.

Western blots were performed with mock-infected FRhK-4 cells as a negative control and HAV-infected FRhK-4 cells, which were treated with various concentrations of hemin for quantifying the viral protein expression. While the HO-1 protein expression was gradually increased with increasing concentrations of hemin, the HAV proteins decreased (Figure 1C). These results showed that increased expression of HO-1 proteins could induce inhibition of HAV proliferation while increasing hemin concentration.

### 3.2. Reduction of the HAV Viral Protein through HO-1 Overexpression

We identified no induction of HO-1 in FRhK-4 cells, which were not treated with hemin (Figure 1C). Therefore, we developed the protein expression vector pcDNA3.1 (+)/mkHO-1 that expressed monkey HO-1. The FRhK-4 cells transfected with mkHO-1 expression vector were infected with HAV 2 days later, and the expression of HO-1 and viral proteins were evaluated using Western blot 3 days later. HO-1 protein expression was identified in the cells transfected with the pcDNA3.1 (+)/mkHO-1 (Figure 2). However, HO-1 expression was not visible in the cells transfected with the empty vector, as expected, and in the HAV-infected cells. A strong induction of HO-1 protein expression by the expression vector caused complete inhibition of HAV replication (Figure 2). These results indicated that increased levels of the HO-1 protein led to less HAV replication.

### 3.3. Inhibition of HAV Caused by Enzymatic Activity of HO-1

To determine whether hemin-mediated inhibition of HAV replication is dependent on HO-1 enzymatic activity, we observed the effect of ZnPP-9, an inhibitor of HO-1 enzymatic activity, on anti-viral activity. FRhK-4 cells were treated with various concentrations of ZnPP-9 simultaneously infected with HAV. Viral RNA and protein levels of HAV were evaluated by qPCR and Western blot. ZnPP-9 could not suppress HAV RNA and protein expression (Figure 3A,B). To further confirm the effect of ZnPP-9 on HO-1 enzyme activity, we treated HAV-infected FRhK-4 cells with ZnPP-9 and hemin. Similar to previous results, hemin treatment could inhibit the expression of both HAV RNA and proteins (Figure 3C,D). However, this inhibition could be partially reversed by ZnPP-9 in a dose-dependent manner (Figure 3C,D). These results showed that HAV inhibition could be mediated by the enzymatic activity of HO-1.

### 3.4. Suppression of HAV Replication by Metabolites of Heme Produced through HO-1 Catalysis

HO-1 catalyzes the generation of CO, ferrous ion, and biliverdin from heme. We evaluated the inhibitory effect of these three metabolites on HAV proliferation. FRhK-4 cells were infected with HAV and simultaneously treated with FeCl_3_, CORM-3, and biliverdin. All three metabolites of heme suppressed the replication of the viral RNA without cytotoxicity (Figure 4A and Appendix A). On the basis of the qPCR results, Western blot assay was conducted using the highest concentrations of FeCl_3_, CORM-3, and biliverdin used in a previous experiment three days after infection. Expression of the VP1 protein of HAV was decreased by the treatment of the three metabolites, consistent with the reduced replication of the viral RNA (Figure 4B). Through these results, we confirmed that the breakdown of heme into CO, ferrous ion, and biliverdin could suppress HAV replication.

### 3.5. Suppression of HAV Replication by Other HO-1 Inducers

We used other HO-1 inducers such as andrographolide and CoPP-9 to confirm that HAV was not suppressed by other unknown properties of hemin. The antiviral activity was determined using andrographolide, which is a non-porphyrin-derived natural product, and CoPP-9, which is another porphyrin-derived HO-1 inducer. Cells treated with the two HO-1 inducers significantly induced HO-1 mRNA, as expected (Figure 5A). When HAV RNA was determined in the same cells, the HAV RNA levels were significantly reduced in a dose-dependent manner (Figure 5B). Finally, we analyzed the viral protein levels by Western blot analysis to verify their antiviral activities. The two HO-1 inducers decreased the VP1 protein of HAV (Figure 5C). Through these further experiments, we confirmed that HO-1 induction by not only hemin but also other HO-1 inducers could lead to inhibition of HAV replication.

## 4. Discussion

In this study, we evaluated HO-1 as an antiviral protein against HAV infection. Translocation of the Nrf2 protein into the nucleus is required for HO-1 induction [23]. Hemin and CoPP-9 are organic porphyrin compounds consisting of four pyrrole rings. These compounds can induce HO-1 by dissociating the Keap1–Nrf2 complex, and the dissociated Nrf2 protein is translocated into the nucleus [21]. Another HO-1 inducer, andrographolide, is a non-porphyrin-derived natural product. It can induce HO-1 through nuclear translocation of Nrf2 by activating p38 mitogen-activated protein kinases (MAPK) and inhibiting glycogen synthase kinase-3β (GSK-3β) by its phosphorylation [27].

On the basis of the above-mentioned findings, we induced HO-1 upregulation by treating FRhK-4 cells with hemin, CoPP-9, and andrographolide and assessed the potential antiviral activities against HAV infection. Hemin was used in this study mainly because it is safer than other inducers and was approved by the Food and Drug Administration [28]. According to the cytotoxicity experiments, the concentration of hemin showing 50% toxicity in FRhK-4 cells was approximately 15.76 times the concentration of hemin that induced 50% HAV reduction. This indicates that the selectivity index is approximately 15.76, which is higher than other FDA-approved chemicals [29]. HO-1 expression was dose-dependently upregulated by the HO-1 inducers and the expression levels of HAV RNA and proteins was inversely reduced. To exclude antiviral effect against HAV infection by other factors of chemical treatments such as the porphyrin effect, we selectively expressed the HO-1 protein in FRhK-4 cells by transfecting a plasmid vector which includes monkey HO-1 gene [30]. We confirmed that this group presented a significant decrease in HAV replication compared to the control group that was transfected with an empty plasmid vector without the HO-1 gene. Additionally, the induction of HO-1 via other inducers such as CoPP-9 or andrographolide was conducted. These chemicals could also elicit the suppression of HAV infection in a dose-dependent manner. To our knowledge, this study is the first to demonstrate that induction and overexpression of HO-1 could suppress HAV replication in FRhK-4 cells.

HO-1 induction exerts antiviral effects through two pathways. One is through the production of metabolites via heme degradation, and the other is through an interaction with interferon regulatory factor 3 (IRF3) to activate the interferon pathway [21]. To determine which activity of HO-1 inhibits the HAV infection, the HO-1 inhibitor ZnPP-9 was used for further experiments. Previous studies have shown that ZnPP-9 induces HO-1 protein expression but inhibits its enzyme activity, which induces the production of CO, ferrous ion, and biliverdin from heme [31,32]. When HAV-infected FRhK-4 cells were treated with ZnPP-9, it significantly reversed the antiviral activity of hemin against HAV replication. However, ZnPP-9 did not suppress HAV replication by itself, which indicates that the enzymatic activity of HO-1 is necessary for HAV suppression, such as in the case of Ebola virus or Pseudorabies virus infection [18,22]. Although ZnPP-9 significantly suppressed the enzymatic anti-viral activity of HO-1, it did not completely inhibit replication of HAV. Similar results were observed in Pseudorabies virus when the virus-infected cells were treated with the ZnPP-9 [18]. The reason for incomplete reversion of the anti-viral activity of HO-1 by ZnPP-9 might be attributed to the residual activity of hemin or other unknown effects of HO-1.

To confirm that HAV infection is suppressed by the enzymatic activity of HO-1, we conducted further experiments using HO-1 downstream products (CO, Fe^3+^, and biliverdin). The products of HO-1 are called ‘therapeutic funnels’ because they exhibit direct cytoprotective effects or help other therapeutic molecules, such as prostaglandins, interleukin-10, and inducible nitric oxide synthase [16]. Additionally, these enzymatic products display anti-viral activities against several viruses [18,22,33]. In our study, treatment with FeCl_3_, CORM-3, or biliverdin, significantly reduced the production of HAV RNA and proteins (*p* < 0.01). Iron is known to suppress the replication of HCV by inhibiting the activity of the non-structural protein RdRp [34]. In this study, we identified the reduction of RdRp gene expression of HAV when hemin and three metabolites of heme were treated to the HAV-infected cells. These results suggested that the direct action of downstream metabolites of HO-1 could inhibit HAV replication. Both the reversion of hemin-induced antiviral effects by ZnPP-9 and induction of antiviral effect by heme byproducts produced through HO-1 activity suggested that the suppression of HAV replication is caused by the activities of induced HO-1.

A study indicated that induction or overexpression of HO-1 could down-regulate pro-inflammatory cytokines and interfere fibrosis progression in liver [35]. Those previous studies and our anti-HAV effects presented in this study altogether imply HO-1 might be used as a new therapeutic measure to hepatitis A.

In summary, the results obtained in the present study suggest that HO-1 is effective for the suppression of HAV replication in vitro, and its enzymatic products appear to contribute to these antiviral activities. In the absence of a specific drug for HAV infection, HO-1-inducer or HO-1-expression drugs would be considered a new treatment strategy against HAV infection. In a further study, the anti-HAV activity of HO-1 would be evaluated in animal models. Based on these in vitro results, the development of optimal animal models and ensuring the safety and effectiveness of HO-1 induction are important to promote the development of new therapeutics.

## Figures and Tables

**Figure 1 pharmaceutics-13-01229-f001:**
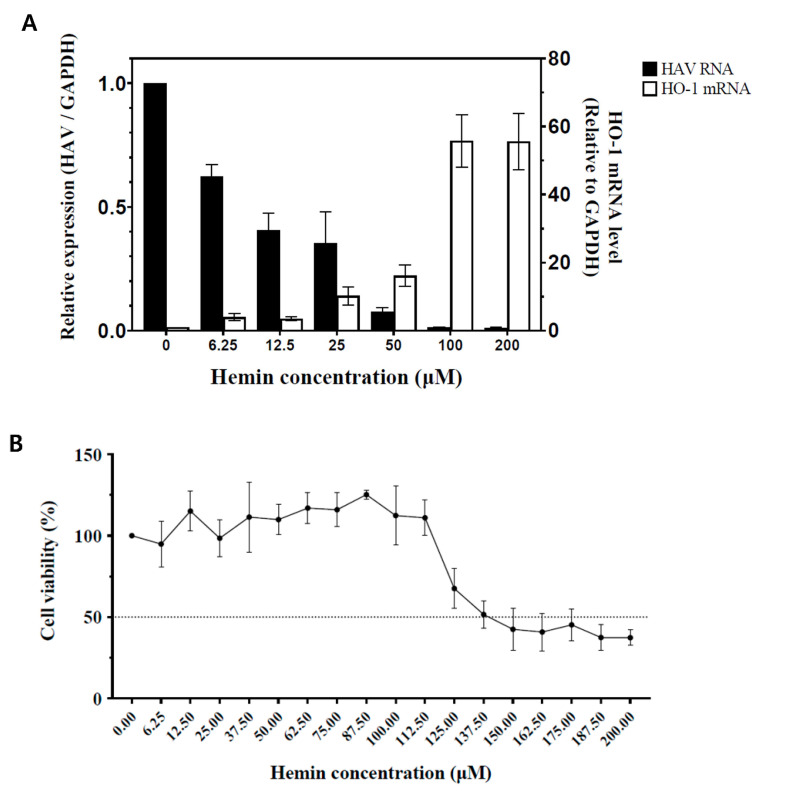
Changes in the amount of HAV RNA and HO-1 induction in FRhK-4 cells according to the concentrations of hemin. (**A**) FRhK-4 cells infected with HAV strain HM-175/18f, clone B (10 GE/cell) were treated for 72 h without or with various concentrations of hemin. As the concentration of hemin increased, the amount of HO-1 mRNA increased as expected, and HAV RNA decreased accordingly. (**B**) Cell viability percentages (compared to hemin-untreated cells) of FRhK-4 cells at 72 h post-treatment with the indicated concentrations of hemin. (**C**) Changes in HO-1 and HAV proteins according to hemin concentration. As the concentration of hemin increased, the HO-1 protein band became thicker and the amount of viral proteins decreased. GAPDH, a housekeeping protein, showed that the same number of cells was used in this experiment. Data are presented as the mean ± standard deviations (SD) of at least 3 independent experiments.

**Figure 2 pharmaceutics-13-01229-f002:**
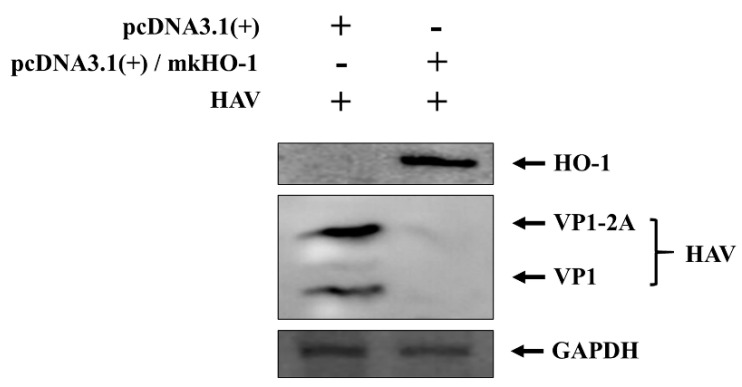
HO-1 overexpression inhibited HAV proliferation. FRhK-4 cells were transfected with pcDNA3.1 (+) vector containing monkey HO-1 gene or empty pcDNA3.1 (+) vector control. At 2 days after transfection, the FRhK-4 cells were infected with 10 MOI of HAV. Western blotting was conducted using cell lysates 3 days after the viral infection. GAPDH was used for proving equal loading volume.

**Figure 3 pharmaceutics-13-01229-f003:**
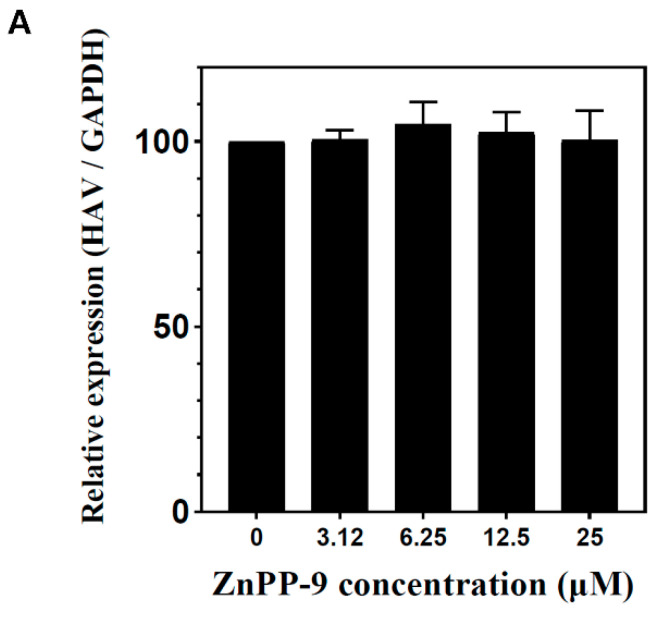
Inhibition of HAV proliferation is due to the enzyme activity of HO-1. (**A**,**B**) ZnPP-9, which induces HO-1 expression but suppresses enzyme activity, could not suppress HAV proliferation. As the concentration of ZnPP-9 increased, the amount of HO-1 protein increased; however, the amount of HAV did not decrease. (**C**,**D**) Inhibition of the HO-1 enzymatic activity by ZnPP-9 prevents the inhibition of HAV proliferation by hemin. In the absence of ZnPP-9, the treatment of hemin reduces the amount of HAV proteins; however, combination with ZnPP-99 reversed the inhibitory effect of hemin on HAV expression. Data are presented as the mean ± standard deviations (SD) of at least 3 independent experiments. * *p* < 0.05; ** *p* < 0.01; *** *p* < 0.001.

**Figure 4 pharmaceutics-13-01229-f004:**
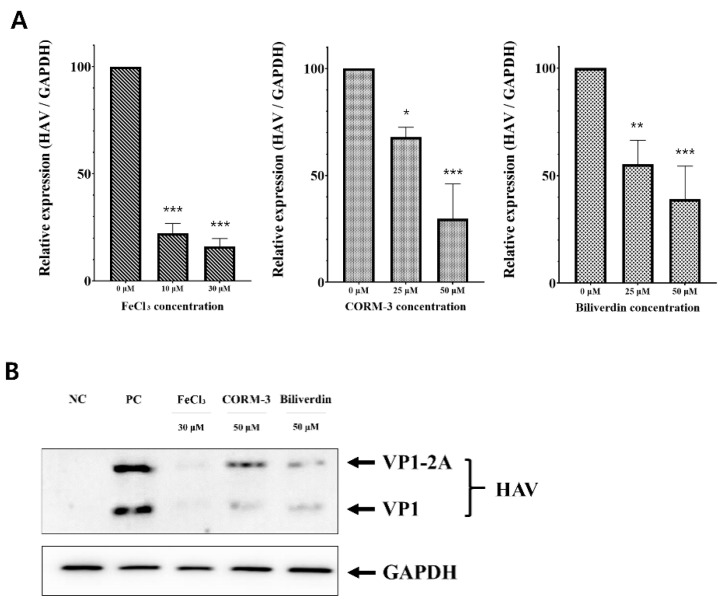
Inhibition of HAV replication by three representative metabolites of heme that are produced through the enzymatic activity of HO-1. HAV replication was suppressed by iron (III) chloride (FeCl_3_) and carbon monoxide-releasing molecule-3 (CORM-3), which induce free iron and CO, respectively, and biliverdin. (**A**) Treatment with FeCl_3_, CORM-3, and biliverdin could significantly suppress HAV RNA replication in the cell. (**B**) The protein level of HAV VP1 decreased when the cells were treated with FeCl_3_, CORM-3, or biliverdin. Data are presented as the mean ± standard deviations (SD) of at least 3 independent experiments. * *p* < 0.05; ** *p* < 0.01; *** *p* < 0.001.

**Figure 5 pharmaceutics-13-01229-f005:**
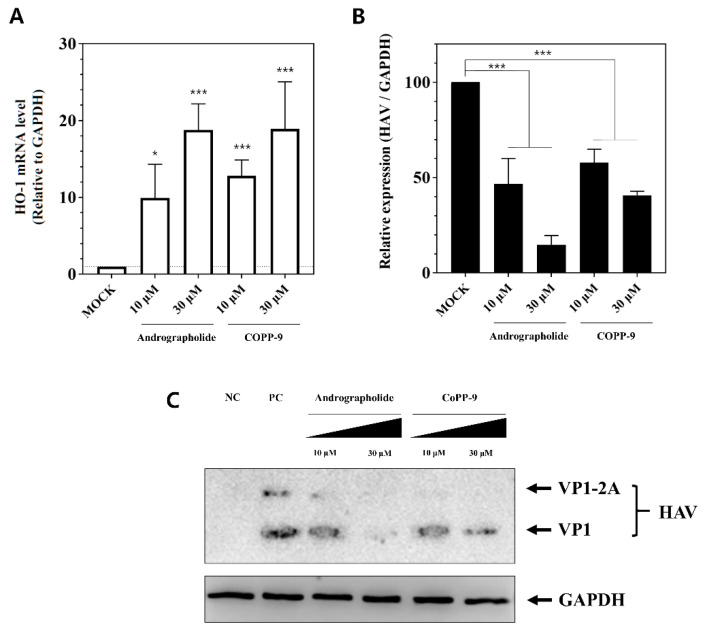
Suppression of HAV replication by other HO-1 inducers, andrographolide and CoPP-9. Non-porphyrin-derived HO-1 inducer (andrographolide) or another porphyrin derived HO-1 inducer (CoPP-9) were used to prove that the inhibition of HAV replication was not caused by other action of hemin. (**A**,**B**) Andrographolide and CoPP-9 also could significantly increase HO-1 mRNA and suppress HAV RNA replication depending on the concentration. (**C**) HAV VP1 proteins decreased depending on the concentration. Data are presented as the mean ± standard deviations (SD) of at least 3 independent experiments. * *p* < 0.05; *** *p* < 0.001.

**Table 1 pharmaceutics-13-01229-t001:** Cytotoxic concentration (CC_50_) and effective concentration (EC_50_) of hemin in FRhK-4 cell line and hepatitis A virus.

Virus	FRhK-4 CC_50_:159.47 μM	
	EC_50_	SI ^1^
HAV (HM-175/18f)	10.12 μM	15.76

^1^ SI: selectivity index (CC_50_/EC_50_).

## Data Availability

Data are contained within the article.

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
