# Peer review of "Heme Oxygenase-1 Exerts Antiviral Activity against Hepatitis A Virus In Vitro"

_pharmaceutics, 2021, doi:10.3390/pharmaceutics13081229_

Round 1

Reviewer 1 Report

The aim of this article was to present the to examination the antiviral effect of  HO-1 on HAV infection in vitro, the induction and expression of HO-1 effectively inhibited  HAV, which implies that a novel and effective strategy could be developed against HAV  infection.

In general, the manuscript is nicely organized and written. I recommend this for publication after minor corrections.

Major concern:

-For line 33 there is a need to a reference for “HAV causes only acute  hepatitis, not chronic hepatitis “

-Revise the abbreviation in line 63

-Please Add for paragraph 2.3 missed the amplification conditions for the RT-PCR cycles

-The paragraph 3.2. should be in the first of the results as 3.1. to make it more clear

-These metabolites treatment  was performed  on FRhK-4 cells transfected with the empty vector??

The mechanism in which HO-1 overexpression decreasing  the viral proteins amount as shown in western blots still unknown, is it due to protein-protein interaction between HO-1 and VP1 or due to interaction between HO-1 and the Viral RNA through RNA binding sites for example, not only the RNA sequence but also the RNA architecture imposed by the presence of specific structural domains mediates the interaction with host RNA-binding proteins (RBPs), ultimately affecting virus multiplication and spreading, like for example the IRES element  upstream of a single ORF encoding a polyprotein, which is cotranslationally and post-translationally processed into the mature viral proteins, there are many ways to complete the virus replication cycle,

hence the three metabolites of heme which founded suppressed the replication of the viral RNA according to RT-PCR are not sufficient to assume that there is  a reduction in the virus replication since the amplified region in RT-PCR is non coding  region and not the RdRp which is the key player in the virus replication, supporting my opinion that they found the iron has been reported to inhibit the activity of the non-structural RNA-dependent RNA polymerase in HCV according to  Fillebeen C, Rivas-Estilla AM, Bisaillon M, Ponka P, Muckenthaler M, Hentze MW, et al. Iron inactivates the RNA polymerase NS5B and sup- presses subgenomic replication of hepatitis C virus. J Biol Chem 2005;280: 9049-9057. So why not it could be the same in HAV?

-I suggest it will be better if you include in this work a RT-PCR for the non structure proteins in the same cell lines in order to assume that the reduction include the major viral genes.

a lot of studies confirmed that some viruses continue to replicate after deletion or splicing by microRNA in 2 or 3 independent region, also they found that 1 single gene of the virus is able to start infectivity https://doi.org/10.1186/s12985-020-01379-x.

Another important aspect to consider is the microRNA and RNAi in the tested cell line, so a screen for present microRNA in cells infected with HAV with the vector and to compare with the cells with the empty vector, since siRNA and VSR silencing suppressor often overlap we can not exclude that maybe the over expression of HO-1 trigger siRNA signal and this signal interfere with the dsRNA of the virus or with one of its non structure protein which could be a silencing suppressor as discovered in HCV Zhou, H., Qian, Q., Shu, T. et al. Hepatitis C Virus NS2 Protein Suppresses RNA Interference in Cells. Virol. Sin. 35, 436–444 (2020). https://doi.org/10.1007/s12250-019-00182-5

Reviewer 2 Report

The authors presented a research paper showing the antiviral activity of HO-1 against HAV. The publication has interesting data, and it is well structured. However, some data need to be added to reach publication quality. 

My major concern is that WB was used as a quantitative measure to determine antiviral activity. It is well-known that WB analyses are semi-quantitative analyses, and required further validation through qualitative analysis. In fact, when you look at figures 3, 4, and 5, it is obvious that with these data is impossible to quantify accurately. Thus, I will strongly suggest that the authors have to develop these analyses using qPCR (you already have the system validated to detect HAV). 

The result showed that HAV replication was suppressed by iron (III) chloride (FeCl3) and carbon monoxide releasing molecule-3 (CORM-3), and biliverdin. are these compounds toxic for the cells? You need to add cytotoxicity curves for these compounds. 

Znp99 is an HO-1 inductor that suppresses HO-1 activity. Based on your results, the effect of HO-1 on HAV is mediated by the activity of HO-1. Thus, why Znp99 could not revert the antiviral activity to basal levels? Why the reversion is partial (Hemin decrease in ~90%, and in the high dose of Znp99 you reverse/increase up to ~30%)? These answers and explanations need to be added to the discussion. 

The authors conclude their ms with the following statement: In the absence of a specific drug for HAV infection, HO-1-inducer or HO-1-expression drugs would be considered a new treatment strategy against HAV infection.

Is there any reference that supports the HO-1 induction as a therapeutic approach?

HAV targets the liver. What is the expression/function of HO-1 in the liver, the cells used in this work re-ensemble this system?

What are the changes in HO-1 after HAV infection?

What are the major functions of HO-1 in the liver, and what implications could be associated with an HO-1 over induction? 

Do you think that further studies are needed before considering HO-1 inducers as a new treatment for HAV?

The answers to these questions should be reflected in the discussion. 

Round 2

Reviewer 2 Report

The authors had addressed most of the concerns.